# Recognizing the Wadi Fluvial Structure and Stream Network in the Qena Bend of the Nile River, Egypt, on Landsat 8-9 OLI Images

**Polina Lemenkova** * and **Olivier Debeir**

Laboratory of Image Synthesis and Analysis (LISA), École Polytechnique de Bruxelles (Brussels Faculty of Engineering), Université Libre de Bruxelles (ULB), Building L, Campus du Solbosch, ULB—LISA CP165/57, Avenue Franklin D. Roosevelt 50, 1050 Brussels, Belgium; olivier.debeir@ulb.be
* Correspondence: polina.lemenkova@ulb.be; Tel.: +32-471-86-04-59

**Abstract:** With methods for processing remote sensing data becoming widely available, the ability to quantify changes in spatial data and to evaluate the distribution of diverse landforms across target areas in datasets becomes increasingly important. One way to approach this problem is through satellite image processing. In this paper, we primarily focus on the methods of the unsupervised classification of the Landsat OLI/TIRS images covering the region of the Qena governorate in Upper Egypt. The Qena Bend of the Nile River presents a remarkable morphological feature in Upper Egypt, including a dense drainage network of wadi aquifer systems and plateaus largely dissected by numerous valleys of dry rivers. To identify the fluvial structure and stream network of the Wadi Qena region, this study addresses the problem of interpreting the relevant space-borne data using R, with an aim to visualize the land surface structures corresponding to various land cover types. To this effect, high-resolution 2D and 3D topographic and geologic maps were used for the analysis of the geomorphological setting of the Qena region. The information was extracted from the space-borne data for the comparative analysis of the distribution of wadi streams in the Qena Bend area over several years: 2013, 2015, 2016, 2019, 2022, and 2023. Six images were processed using computer vision methods made available by R libraries. The results of the *k*-means clustering of each scene retrieved from the multi-temporal images covering the Qena Bend of the Nile River were thus compared to visualize changes in landforms caused by the cumulative effects of geomorphological disasters and climate–environmental processes. The proposed method, tied together through the use of R scripts, runs effectively and performs favorably in computer vision tasks aimed at geospatial image processing and the analysis of remote sensing data.

**Keywords:** Africa; k-means clustering; image processing; remote sensing; unsupervised classification; programming; visualization; modeling; computer vision; cartography

## 1. Introduction

### 1.1. Background

Satellite images are frequently used as a source of geoinformation for the mapping of land cover types and recognizing land features. Examples of such applications include thematic maps made using the classification of satellite images; these include, for instance, maps of mineral resources [1–3], environmental assessment [4,5], land cover/land use classification [6,7], geologic analysis [8,9], and urban mapping [10]. The success of the

remote sensing data in cartographic processing and mapping is based on the effectiveness of the satellite imagery as a source of information for the recognition of diverse land features and processes.

Among the advantages of using the high-resolution space-borne data as a source of information, one can mention the improved quality of spatial visualization [11,12], and the enabled access to remotely located places and areas otherwise inaccessible, such as deserts [13]. For such places, satellite imagery presents a valuable source of information. Apart from spatial access, satellite images facilitate the temporal analysis of environmental changes or disaster monitoring using the comparative analysis of several scenes. For instance, the comparison of several satellite images covering a target region in different time periods enables us to perform time series analysis, which can be used in detecting climate and environmental changes or monitoring disaster events. A similar image-to-image comparison can also help in assessing the risks of hazards, and to perform ecological surveillance of the target study area using multi-temporal data analysis [14–16].

Applications of remote sensing data in Earth sciences include monitoring desertification and land degradation [17,18], the analysis of deforestation [6], the evaluation of salinization [19] and soil erosion [20], hydrological modeling [21] and the quantification of the flooded areas [22,23], and many more. The spectral patterns of the multispectral Landsat images are especially widely used in geosciences as a source of spatial information for land cover mapping and the recognition of land features [24]. To this end, the geometric, radiometric, and spectral precision of the Landsat scenes can be enhanced through a combination of the high-resolution panchromatic and multispectral channels by fusion methods [25].

Natural resource monitoring benefits from the use of satellite images, which provide a powerful source of information for detecting environmental processes [26]. Furthermore, identifying lithologic land surface structures and extracting extracted structural lineaments is possible using computer vision techniques [27–29]. For instance, research on mineral exploration can be supported using the classified satellite images to discriminate lithological units on the color composites of the images in various ratios [30]. For geomorphological modeling, the Shuttle Radar Topography Mission (SRTM) Digital Elevation Model (DEM) can be overlaid with satellite images [31]. Such data integration enables us to identify the morphometric parameters of the river catchment and drainage area, and to examine the spatial distribution of the agricultural fields with regard to the fluvial networks [32].

Information derived from the classified satellite images enables us to perform the assessment of geomorphological and geological risks to define maps of hazards. This includes the problem of morphometric modeling using remote sensing data for the identification of geologic hazards, as widely investigated in the cartography, geoinformatics, and Earth science communities. Hamdan and Khozyem [33] applied the Advanced Spaceborne Thermal Emission and Reflection Radiometer (ASTER) data and Geographic Information System (GIS) techniques to delineate drainage networks in the Wadi El-Mathula watershed in the Central Eastern Desert. Various methods of geomorphometric modeling exist and can be applied to detect the spatial patterns of fluvial systems, or for hydrogeological modeling aimed at the analysis of groundwater potential [34–36]. The geometry and morphology of the relief and the characteristics of topographic parameters such as slope, aspect, and hillshade can be extracted as morphometric characteristics for the analysis of watersheds using 3D mapping [37].

Extracted information from remotely sensed data can be used for complex hydrological analysis—for instance, predicting wadi flash floods using morphometric parameters that represent the topographic and drainage characteristics of the basins using GIS [38]. Further, such information can be used for the analysis of wadi networks to define basin boundaries and drainage networks, to evaluate the stream channel density, and to model slope and flow directions. Moreover, it is useful for the retrospective modeling of the hydrological processes in the past, to better analyze the geologic setting in the present, using robust

remote sensing data. Such an approach enables deeper insights into the structure of the land surface visible from the space-borne data [39,40].

Moreover, the essential geographic information collected from satellite imagery is useful for operative flash flood monitoring, through integrating data on physiographic features. Other applications of remote sensing data include modeling lithological, geological, and structural features [41], mapping mineral resources [42], and detecting wadi channels based on the combination of the topographic maps and satellite images [43].

*1.2. Motivation and Objectives*

In view of the advantages of satellite images for Earth and environmental studies, the question of the most effective methods of processing these data is arising. A variety of existing works have explored the use of GIS for remote sensing data analysis. The use of ENVI GIS [44–46] well illustrates the traditional methods of remote sensing data processing by the use of supervised classification. More examples in existing studies have used a combination of software for satellite image processing. A prominent example of such an approach is presented by [47], wherein the use of several GIS is applied to the relevant tasks in image processing: ArcGIS for the interpretation of the geological lineaments, Erdas Imaging for data preprocessing, and ILWIS GIS and ENVI GIS to create band color composites and for information extraction. Moreover, the integration of ArcGIS and ENVI GIS is presented in [48,49] for the processing of aeromagnetic and geologic data to map the structural complexity and mineralization in the Southern Eastern Deserts of the Upper Egypt.

Other studies have used Erdas Imagine in processing multi-temporal remote sensing data, e.g., [50,51] monitored land cover changes and computed vegetation indices. Similarly, Ref. [52] presented the application of the Erdas Imaging for geological analysis through identifying rocks by their spectral signatures. Ref. [53] applied ArcGIS for detecting lineaments on DEM and Sentinel-2 images for the geologic analysis of Southern Sinai; Ref. [54] combined ArcGIS and Erdas Imagine for multi-temporal image processing aimed at environmental impact assessment. Such methods correspond to the aim of our approach in terms of using satellite image processing in the monitoring of landscapes of Egypt; however, they are largely tied to specific niches and problem formulations, such as land cover changes, stratigraphic and petrographic analysis, and identifying rocks by the use of remote sensing data. Nevertheless, while these algorithms benefit from the use of the conventional interface in the traditional GIS software, they are limited by the largely restricted functionality of all these programs.

In contrast to these and similar existing studies, here, we propose an R-based use case of satellite image processing to address the problem of using computer vision for extracting geoinformation from remotely sensed space-borne data. The main motivation of this work is to demonstrate that land surface types can be extracted from a short time series of multispectral Landsat 8-9 OLI/TIRS images by k-means clustering, using the libraries of the R language. This work reports an empirical comparison of the classification of the six Landsat scenes for years 2013, 2015, 2016, 2019, 2022, and 2023 to detect changes in the identified land surface types for the target study region of the Qena Bend, Upper Egypt. Furthermore, the results report a comparison of the detected classes by Landsat bands in the false color composite (5-4-3) channels corresponding to the Near-Infrared, Red, and Green bands in the Landsat OLI/TIRS images). The results of the performed classification support the argument that satellite images can be integrated to achieve recognition of the land surface types in the complex drainage network of Wadi Qena, Upper Egypt, and this paper demonstrates a practical approach to achieve satellite image processing using scripts written in the R programming language.

## 2. Prior Work

The Qena Bend is one of the most notable morphological features of the Nile River in Upper Egypt, with a unique, characteristic curvature in its geometry, as shown in Figure 1.

The structural geomorphology of the Qena Bend largely reflects the general geologic setting of the Nile River and the major processes of its evolution [55,56]. The Valley of the Nile River was cut during the late Miocene of the Neogene period and formed later on as a result of the regional and local tectonics. The Qena Formation includes gravel deposits with limestone cobble and heavily weathered soil typical for old terrace deposits [57]. The remaining sediments, outcrops, and paleospecies in fossils have been identified from the Mesozoic Era (Cretaceous period) [58,59] and Paleogene (Paleocene and Eocene epochs) [60,61]. They are reflected in the Cenozoic sediments accumulated in the lacustrine–alluvial fans around the Qena Bend due to the Nile River's evolution in the Neogene; see Figure 2.

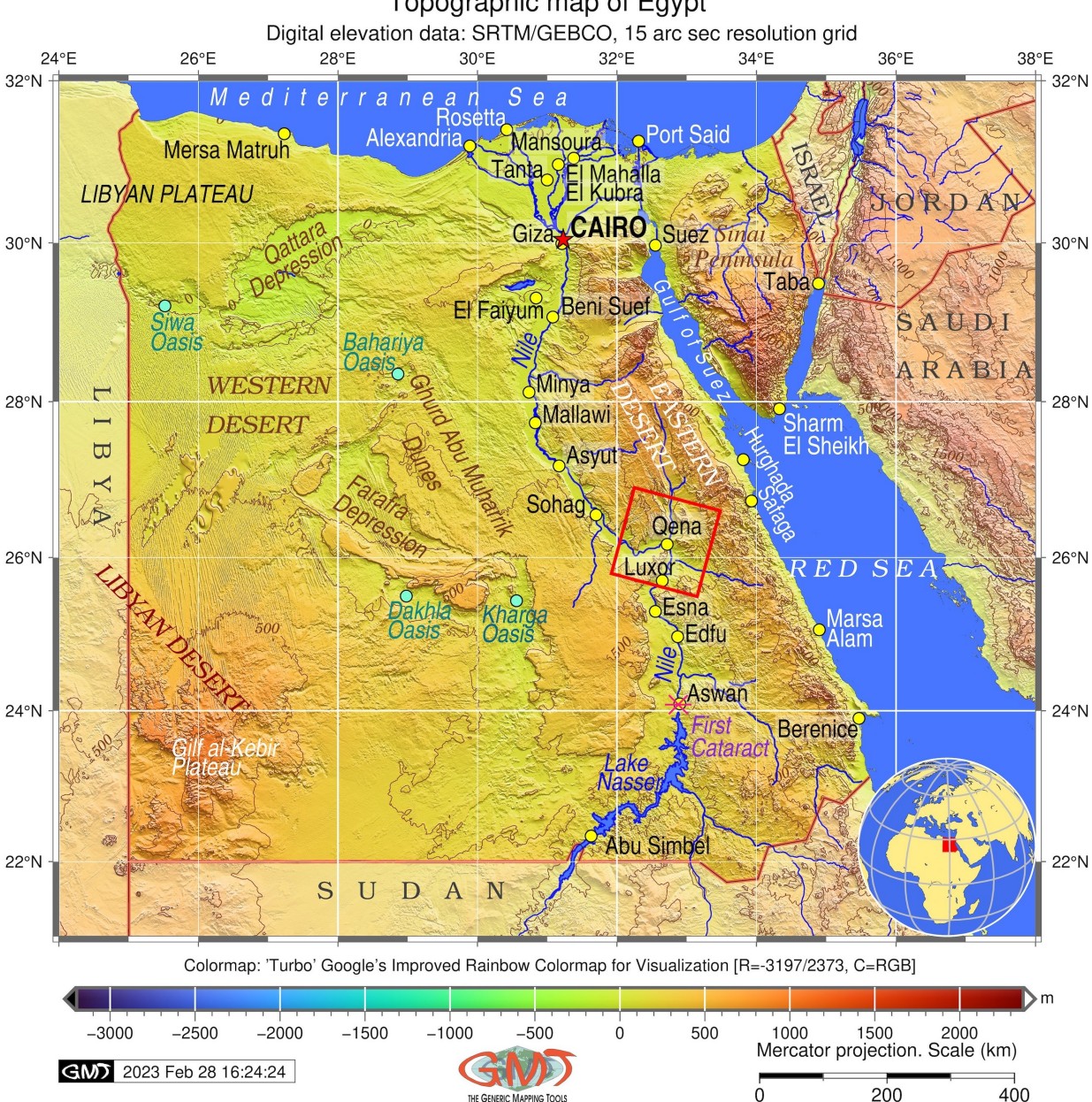

**Figure 1.** Topographic map of Egypt. Target study area of the Qena Bend region is shown in red rotated square. Software: GMT v. 6.1.1. Data source: GEBCO/SRTM. Cartography source: authors.

Apart from the geology and hydrology of the Nile River, the region of the Qena Bend is influenced by the presence of the Eastern Desert, which has a unique climate. Its major features include occasional extreme flash floods—one of the most severe natural disasters. Flash floods occur annually, mostly during spring and autumn, and are caused by torrential

rainfall followed by the disastrous surface runoff thereafter [62]. Such climate processes, aggravated by an interrelated complex of fluvial networks in the Wadi Qena catchment area, as briefly explained above, present favorable conditions for disastrous flash floods. During such events, a complex of the ephemeral rivers that is dry during most of the year is being filled with water from heavy showers and rains [63].

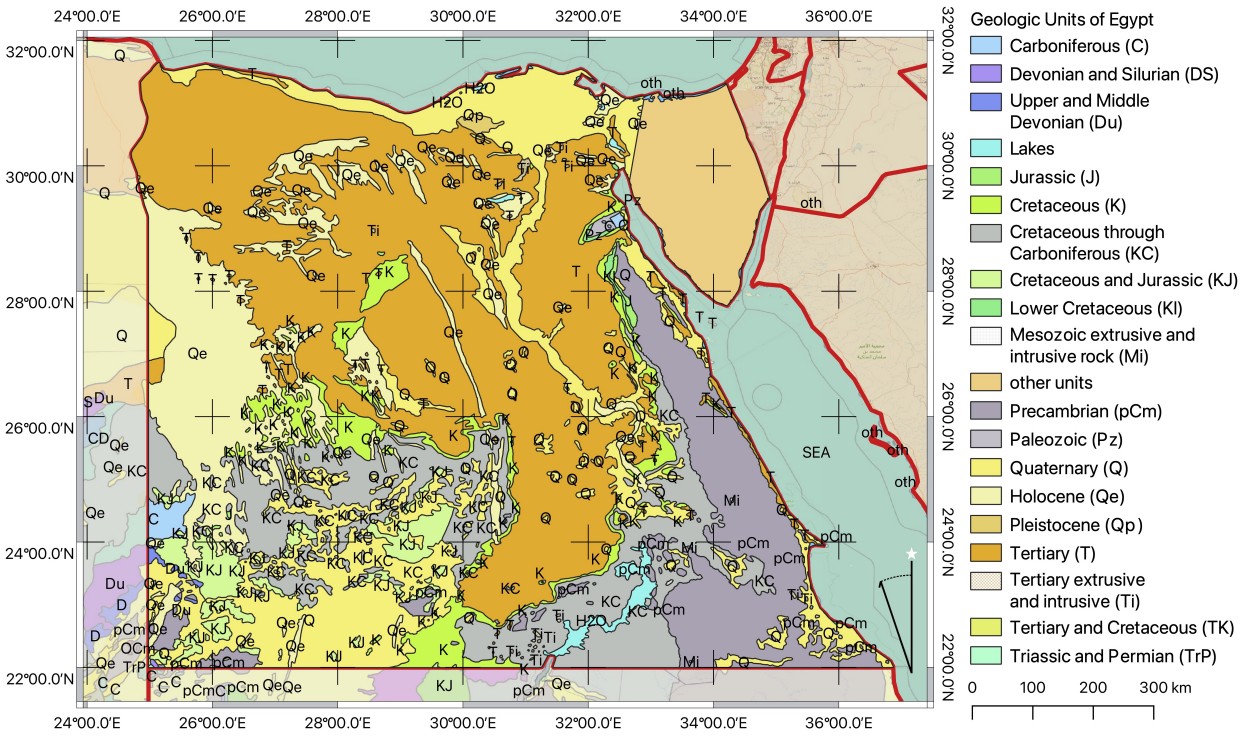

**Figure 2.** Geologic units of Egypt and the Nile River. Software: QGIS v. 3.2.2. Data source: USGS. Cartography source: authors.

The cumulative effects from the wadi network and the arid and semi-arid climate of the Eastern Desert region result in occasional flash flood events and irregular sand dust storms with different durations and intensities. Flash floods occur in diverse regions across Egypt, including Upper Egypt with the Eastern Desert [64,65], the Southern Red Sea Coast [66], the Sinai Peninsula [67], and the Gulf of Suez [68]. The social consequences of flash floods include damaged infrastructure, transport routes, and buildings, and occasional victims [62]. El-Magd et al. report the catastrophic events of flash floods in the Qena governorate recorded from three consecutive events in 2014–2016 [69].

As a result of the geologic evolution, the geomorphology of the Qena Bend region presents a plateau largely dissected by numerous valleys of wadi and geologic lineaments, which dissect the cliffs of the Nile River and the basement plateau composed of the limestone and chalks [70]. This results in a specific geomorphic pattern in the Qena Bend's surroundings, which is notable for its complex drainage network of wadi aquifer systems [71–73]. This drainage network represents a typical feature of the Eastern Desert, which consists of numerous fluvial basins that drain the rainwater towards the Nile River or the Red Sea [74].

The major geomorphic structure of the Nile Valley follows the faults oriented parallel to the Red Sea along the central current of the Nile and the Gulf of Suez in the north [75,76], with the relics of the pre-late Miocene streams remaining in the fluviatile sand and gravel sediments of the Nile delta in Lower Egypt [77]. The discovery of paleorivers in the Sahara by radar imaging systems revealed the orientation of the paleochannels and their flow directions [78]. During the Neogene, it is argued [79] that the ancient Qena Lake broke up

the Nile River's course, which might have affected its present form and specific geometric curvature; see Figure 3.

The occurrence of flood hazards in Upper Egypt points at the significance of the application of geoinformation for the visualization and mapping of regions prone to flooding. The remote sensing data and advanced methods of spatial data processing present robust approaches to the evaluation of relief and the extraction of information on relief and topographic characteristics from satellite images using the methods of computer vision. Such approaches can be used, for instance, for vulnerability mapping or the identification of areas with high geomorphological hazards and exposure to floods [80].

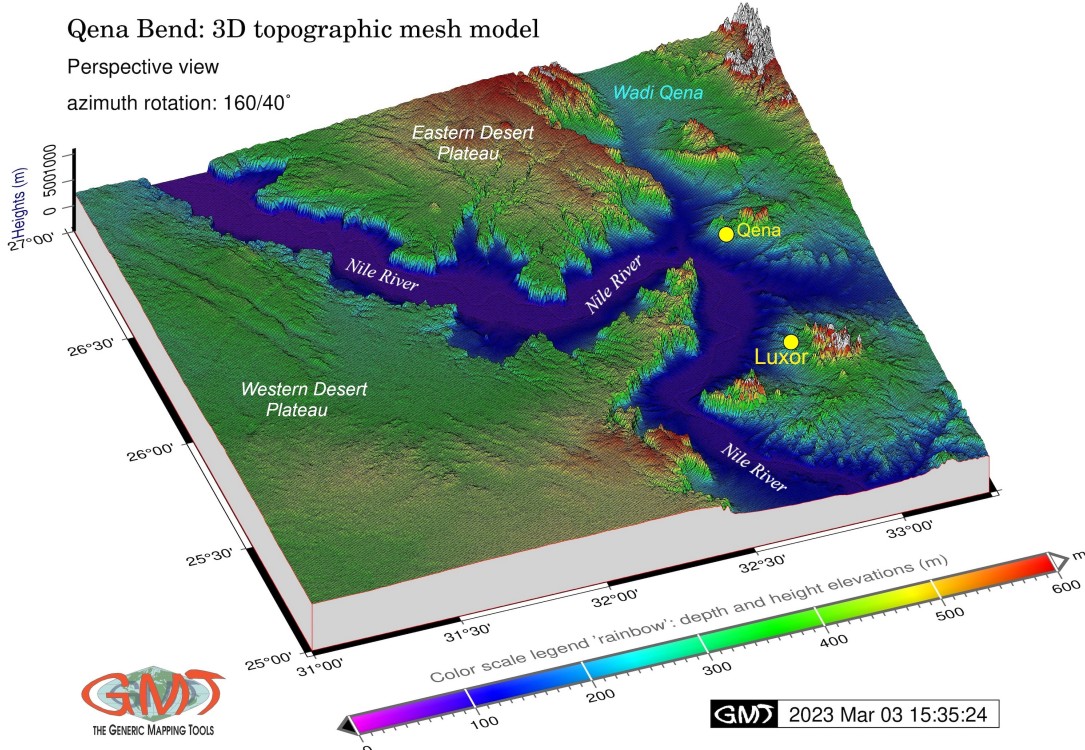

**Figure 3.** Enlarged fragment of the 3D perspective view of Qena Bend, Nile River, Upper Egypt. Software: GMT v. 6.1.1. Data source: GEBCO/SRTM. Cartography source: authors.

## 3. Materials and Methods

### 3.1. Data

In this paper, we present an approach to represent, process, analyze, and compare satellite images, both as classified maps and as extracted information on land cover types and their changes as statistical data derived from the images and summarized in tables. We demonstrate how an R-based approach, utilizing k-means clustering and the spectral reflectance characteristics of the remotely sensed data, can be used to learn how landscapes in Southern Egypt gradually change over time.

Multispectral satellite images such as Landsat TM/ETM+ and OLI/TIRS) provide a valuable source of remote sensing data widely used in environmental studies of Egypt [81–83]. The reason for selecting the Landsat OLI/TIRS images is due to their open-source availability, regular global coverage, and data robustness. The Landsat 8-9 OLI/TIRS scenes have a 30 m resolution in multispectral channels along a 185 km swath for each image (Bands 1 to 7, used for color composites); they are systematically presented, geometrically and radiometrically calibrated, and georeferenced using the World Geodetic System 1984 (WGS84) data, UTM coordinate system (Zone 36 for this case), and terrain corrected by the source provider. Such characteristics of the Landsat 8-9 images make them a reliable and robust source of geoinformation.

Six Landsat OLI/TIRS scenes covering the study area with minimal cloud coverage were selected for the analysis and interpretation of the land cover types. The selection of different image time intervals is explained by the availability of the Landsat 8-9 OLI/TIRS with cloud-free scenes covering the target area. In particular, the Landsat OLI/TIRS sensor is improved against the earlier versions of Landsat products in terms of technical and spectral characteristics, such as narrower spectral bands, a refined radiometric resolution, and better noise detection and calibration parameters [84]. Since the spectral characteristics of the older Landsat products (Landsat TM and Landsat MSS) contain worse parameters, we only selected the data from the Landsat OLI/TIRS with the first images available starting from 2013, when the first OLI/TIRS sensor was launched.

Given the advantages of the Landsat 8-9 OLI/TIRS sensor, which provides moderate-resolution scenes with multispectral bands available for the dates between 2013 and 2022, they were selected for this research. The time span was selected based on the availability of the data in the Qena area that demonstrate low cloudiness in the winter period with an interval of 1–3 years. The images were selected from the winter period to avoid impacts of the arid climate on vegetation parameters. Thus, the images were acquired in November for the Landsat-8 scenes for years 2013 to 2022 and January for Landsat-9 images for 2023. The technical characteristics and metadata of the Landsat satellite images used for image processing are summarized in Table 1.

**Table 1.** Satellite images used for computing the VIs: Landsat-8 OLI/TIRS collected from the USGS [1].

| Date | Spacecraft | Landsat Product ID | Scene ID | Cloudiness |
|---|---|---|---|---|
| 16 November 2013 | Landsat 8 | LC08_L2SP_175042_20131116_20200912_02_T1 | LC81750422013320LGN01 | 0.89 |
| 22 November 2015 | Landsat 8 | LC08_L2SP_175042_20151122_20200908_02_T1 | LC81750422015326LGN01 | 0.05 |
| 8 November 2016 | Landsat 8 | LC08_L2SP_175042_20161108_20200905_02_T1 | LC81750422016313LGN01 | 0.31 |
| 17 November 2019 | Landsat 8 | LC08_L2SP_175042_20191117_20200825_02_T1 | LC81750422019321LGN00 | 1.16 |
| 9 November 2022 | Landsat 8 | LC08_L2SP_175042_20221109_20221121_02_T1 | LC81750422022313LGN00 | 0.02 |
| 20 January 2023 | Landsat 9 | LC09_L2SP_175042_20230120_20230122_02_T1 | LC91750422023020LGN01 | 0.32 |

[1] The sensor ID is common for all the scenes: Landsat 8-9 OLI/TIRS (Operational Land Imager and Thermal Infrared Sensor), Collection 2 Level 2. Path/row parameters common for all the images: 175/42. Coverage: Qena Bend, the Nile River, Upper Egypt. Image source: the USGS.

Besides the Landsat OLI/TIRS images, we also use the topographic and geologic grids to map the study area and analyze the geomorphology in 2D and 3D formats. The interpretation of these ancillary data helps to analyze and verify the impact of the geologic processes and lithological structure in the temporal variations in land cover and surface structure. In such a way, this work is based on the use of two major types of data: six satellite Landsat 8-9 OLI/TIRS images and topographic–geological datasets; see Figure 4. Various land surface objects have different spectral reflectance corresponding to the Digital Numbers (DN) of pixels within a given wavelength interval, which corresponds to different bands (or channels) of the Landsat images. Such an important feature of the objects visible on the images enables their identification by computer vision. This is made possible using the recognition of their characteristics through the classification of Landsat bands taken as RGB triplets of band combinations. In this study, we used different color composites for the visual analysis of the scenes and false color composites for k-means clustering.

The specific geometric curvature of the Nile River in the Qena Bend area is perfectly visible in the Landsat images in natural colors; see Figure 5. Here, it is possible to discriminate the floodplain of the Nile River in a bright green color and the dark blue thread of the river itself contrasting against the surroundings of the sands of the Western and Eastern Deserts. The areas of dark crimson represent massifs of bare soil, while the ivory color indicates a complex wadi network with its typical dendritic pattern contouring the valleys, as visible in Figure 5. To further support our analysis, we integrated geological, geomorphological, and climate data from prior works. High-resolution topographic data were used for the inspection of the relief in the region and to visualize the geomorphology

in 2D and 3D modes, while the remote sensing data were used for the comparative analysis of changes in the wadi channels in the Qena Bend area and surroundings, both by image processing and k-means clustering algorithms using R [85].

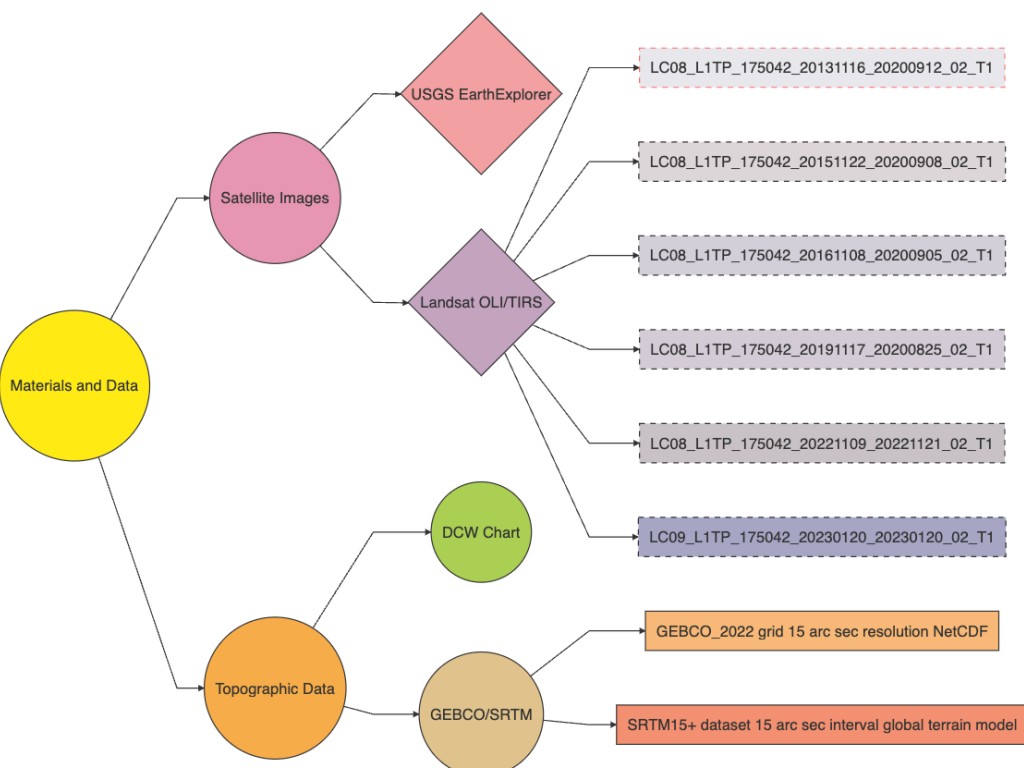

**Figure 4.** Data sources: summary of the materials used in this study. Software: R version 4.2.2, 'Mermaid' library. Flowchart source: authors.

The integration of such data, as well as the analysis of the geological, geomorphological, and climate setting based on prior works, was used to evaluate the variations in land cover types illustrating the risks of hazards in the Qena Bend area. By combining and analyzing these data, we were able to evaluate variations in land cover types and illustrate the hazard risks in the Qena Bend area. Similar examples of data integration for hydrogeological modeling are presented in earlier studies focused on evaluating the aquifer potentiality in the Eastern Desert [86,87], 2D and 3D modeling for land surface slope gradient assessment [88], and the recognition of structural lineaments and fluvial channels and catchment analysis [89]. These studies all demonstrate approaches to data fusion that benefit from advancements in computer vision algorithms for the extraction of information from remote sensing data [90].

### 3.2. Methods

3.2.1. Research Concept and Advantages

The concept of the performed research includes data selection and collection from the USGS repository, data processing by clustering methods, analysis, and interpretation of the obtained results. After data capture, cartographic visualization, and image preprocessing, clustering is the major step in the research workflow. The essential approach of clustering is that it partitions the dataset into several clusters (or groups) of pixels using algorithms of data partition. Diverse types of clustering techniques exist in data analysis, with their own strengths and weaknesses, due to the complexity of information [91]. Some examples of clustering include the algorithms that identify centroids, density, or distribution in a dataset, which enable us to split the data into several groups according to the distance of

each particular pixel to the center of the cluster. Currently, clustering techniques are widely used in programming tools including machine learning [92].

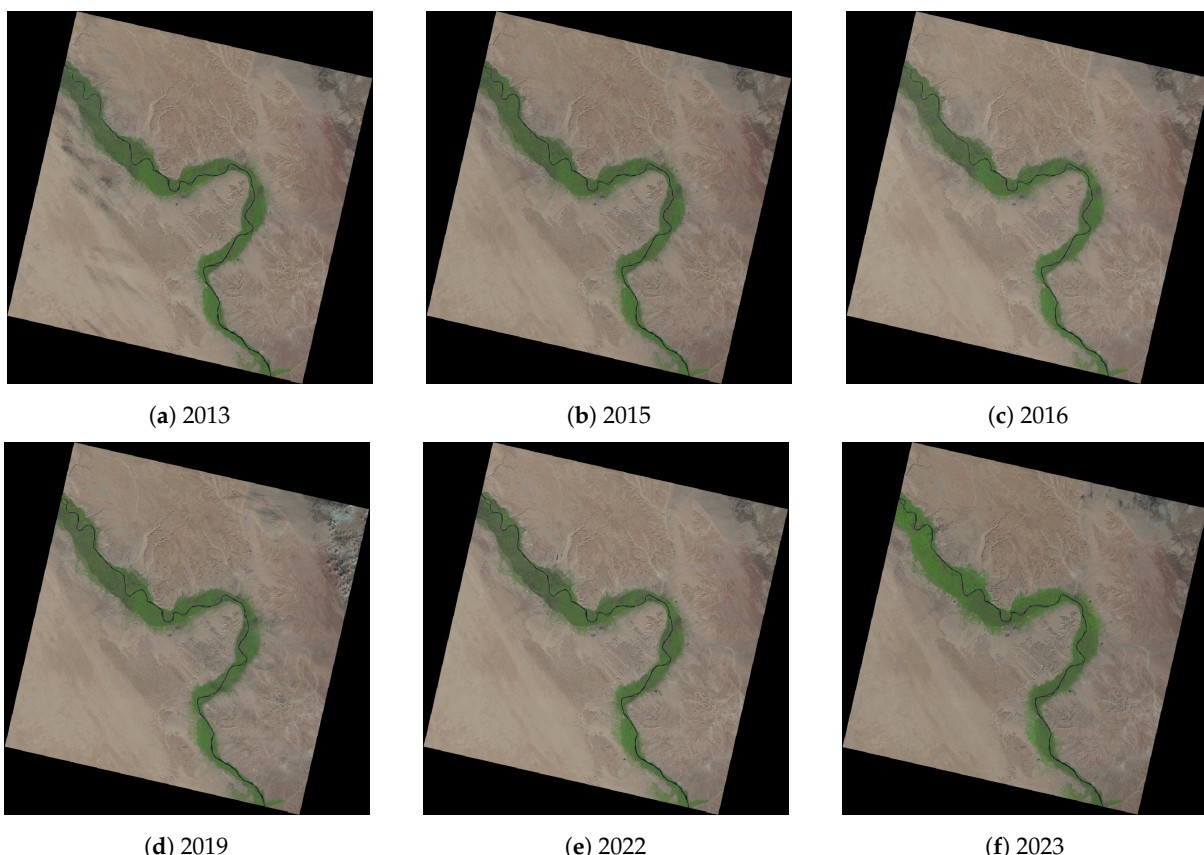

**(a)** 2013          **(b)** 2015          **(c)** 2016

**(d)** 2019          **(e)** 2022          **(f)** 2023

**Figure 5.** Qena Bend of the Nile River, Egypt, visible on Landsat 8-9 OLI/TIRS images in natural RGB color for six years: (**a**) 16 November 2013, (**b**) 22 November 2015, (**c**) 8 November 2016, (**d**) 17 November 2019, (**e**) 9 November 2022, (**f**) 20 January 2023.

In spatial data analysis, clustering is used for image analysis as a tool for unsupervised classification. A clustering algorithm separates pixels in the image into several groups (or clusters) according to their spectral signatures and assigns these pixels to the defined clusters. The most well-known algorithm for clustering is k-means. Other examples include hierarchical clustering [93,94], partition clustering [95], fuzzy clustering [96], mean shift [97,98], density-based clustering [99,100], model-based clustering [101], and Density-Based Spatial Clustering of Applications with Noise (DBSCAN) [102–104].

In this study, we used the k-means algorithm of R, which defines clusters of pixels in the image based on their similar features and is embedded in the R programming language [105]. The concept of k-means clustering includes the iterative process of evaluation of pixels' values, wherein each point is assigned to a specific cluster group corresponding to a defined number of clusters. This approach is straightforward, implemented by the 'raster' package, and accurately classifies the data in the satellite image. k-means performs the simplification of the large massifs of pixels in the image by partitioning them into groups (e.g., land cover types). The advantages of the k-means concept include easy adaptability to new datasets and a relatively straightforward algorithm of implementation in R, which enables us to process the dataset in a few seconds.

### 3.2.2. Data Processing Workflow

The 2D and 3D maps showing the topography and geomorphic structure of the Wadi Qena region were prepared using the Generic Mapping Tools [106] via the existing methods

of cartographic scripting [107]. The general workflow in R is summarized in Figure 6, with the listed libraries, and included the following steps of data processing.

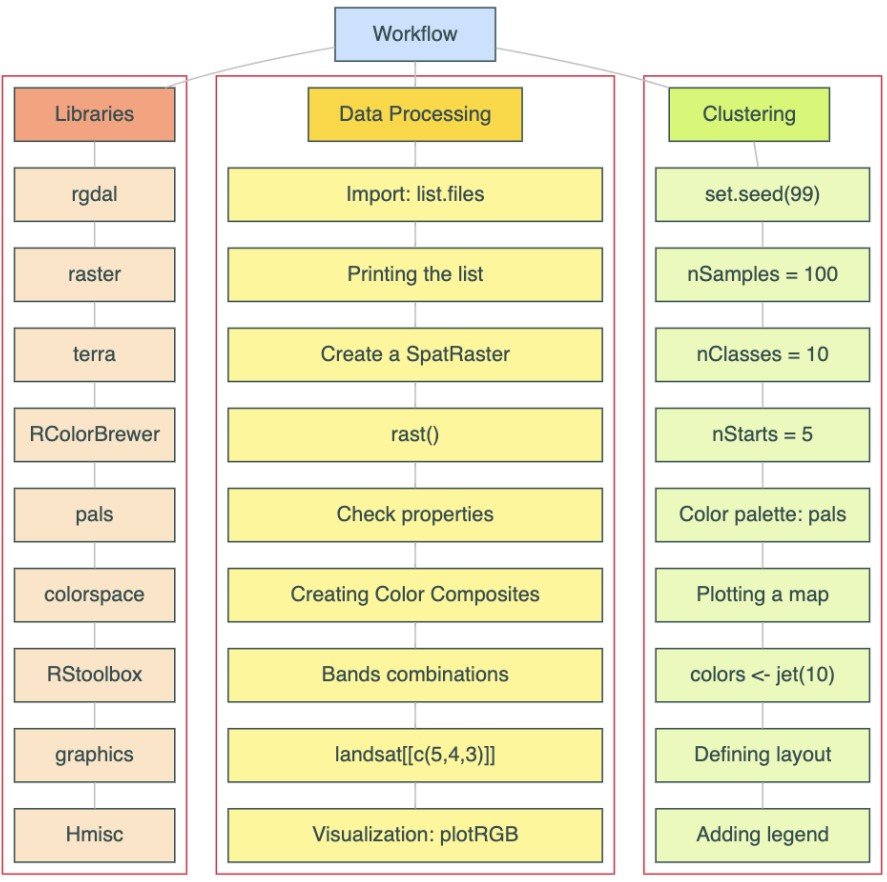

**Figure 6.** Methodological workflow of the processes and research steps applied in this study. Software: R version 4.2.2, 'DiagrammeR' library. Flowchart source: authors.

Specifically, the workflow included the following steps (the examples below are given for the image from 2013 and repeated for all the images, respectively). First, the images were read in to the RStudio environment as a stack of TIFF images (.tif files) using the R command *Landsat2013 <- list.files()*. A subset list of the file names was generated within a working folder on the computer. Then, the 'SpatRaster' object, which reads the raster data's spatial structure, was created as follows: *landsat <- rast(Landsat2013)*. The band designations for the Landsat RGB triplets were performed using command *landsatRGB <- landsat[[c(5,4,3)]]* (here, the case of false color composites).

Afterwards, the image was visualized using the command *plotRGB(landsatRGB, r = 1, g = 2, b = 3, axes = FALSE, stretch = "lin")* of the 'raster' library. The classification was based on the k-means clustering algorithm. The classification was run using the following command: *unC <- unsuperClass(landsatRGB, nSamples = 100, nClasses = 10, nStarts = 5)*. Here, the number of classes was defined as 10, for good separation of the land cover classes in the Wadi Bend region. We also considered a higher number of classes; however, they resulted in more coarse results. For instance, 5 and 7 classes merged several land cover classes that were evidently different, while 20 classes, in contrast, presented unnecessary partitioning of the image into subclasses that should be jointed. The color palette was defined using the 'RColorBrewer' library by *colors <- jet(10)*, and the visualization of the image was performed using the command *plot()*. The legend was then added using the command *legend()*.

The triplets of Bands 5 (Near Infrared), 4 (Red), and 3 (Green) were selected for the following reasons. Healthy vegetation has high absorption in Red (Band 4) and high reflectance in Green (Band 3) and NIR (Band 5). Therefore, such a combination helps to

better discriminate vegetation areas in the floodplain of the Nile from soil, sandy, and desert areas in the Western and Eastern Deserts, as well as the wadi valleys. In contrast, soil has lesser reflectance spectra in NIR because its reflectance changes along with the wavelength, which is why such a combination of bands was used instead for the calculation and clustering of the pixels of various land cover classes based on their distinct spectral signatures. Additionally, all the maps derived from the clustering of the satellite images were georeferenced automatically with metadata identified by R library 'raster': WGS84 datum, UTM coordinate system Zone 36.

Nevertheless, the spectral signature of the land surface differs according to the soil and rock types. The geochemical properties of soil are reflected in the presence of organic carbon, the moisture percentage, the level of salinity, and the relevant environmental characteristics of agricultural land (content of sulfides, fertilizers, pesticides, etc.). Likewise, the reflectance spectra of diverse rock types differ significantly, which enables us to discriminate them on the satellite images using algorithms of computer vision. Therefore, we considered various combinations of the Landsat OLI/TIRS bands for the analysis of land surface types in the images, as shown in Figures 7–9.

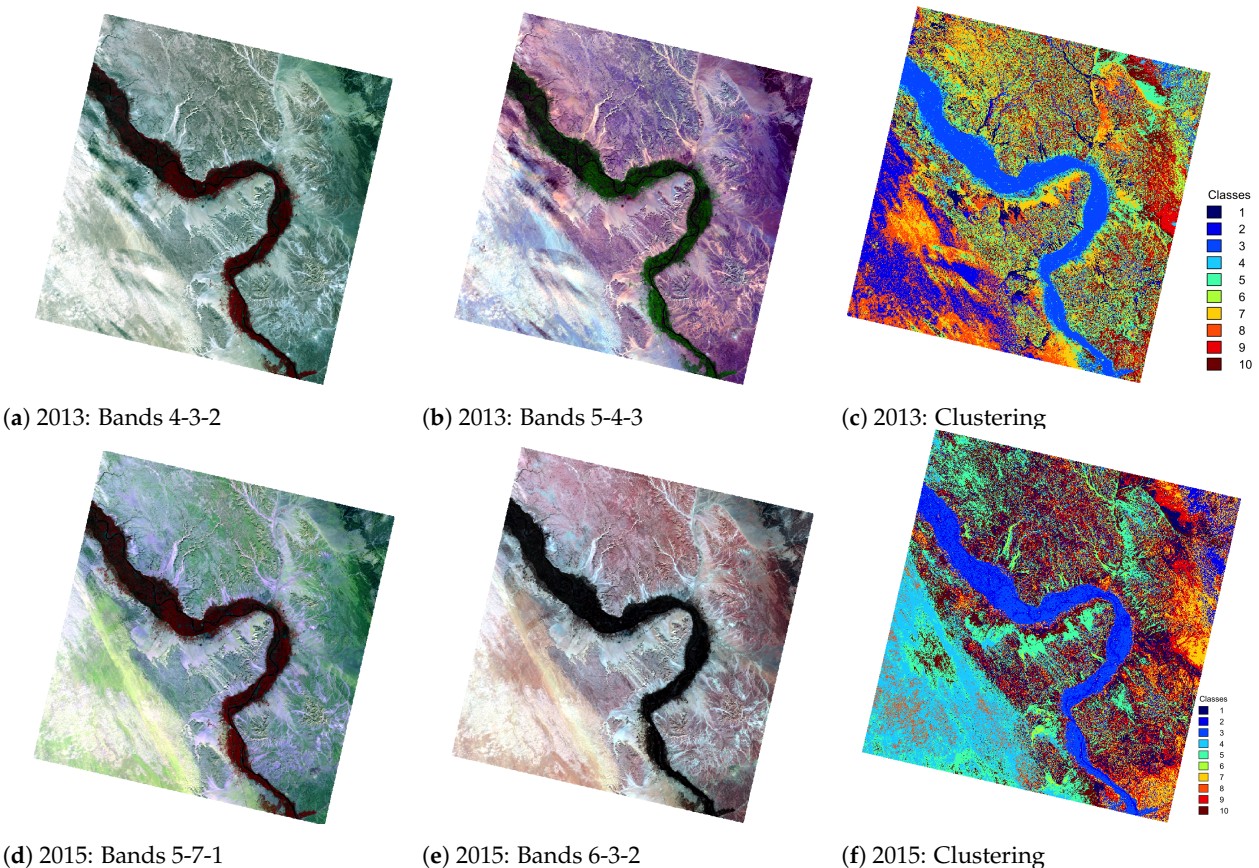

(**a**) 2013: Bands 4-3-2    (**b**) 2013: Bands 5-4-3    (**c**) 2013: Clustering

(**d**) 2015: Bands 5-7-1    (**e**) 2015: Bands 6-3-2    (**f**) 2015: Clustering

**Figure 7.** Qena Bend of the Nile River: false color composites and clustering of the Landsat 8-9 OLI/TIRS images: (**a**) 2013 RGB in 4-3-2 Bands, (**b**) 2013 RGB in 5-4-3 Bands, (**c**) 2013 unsupervised classification, (**d**) 2015 RGB in 5-7-1 Bands, (**e**) 2015 RGB in 6-3-2 Bands, (**f**) 2015 RGB: Clustering.

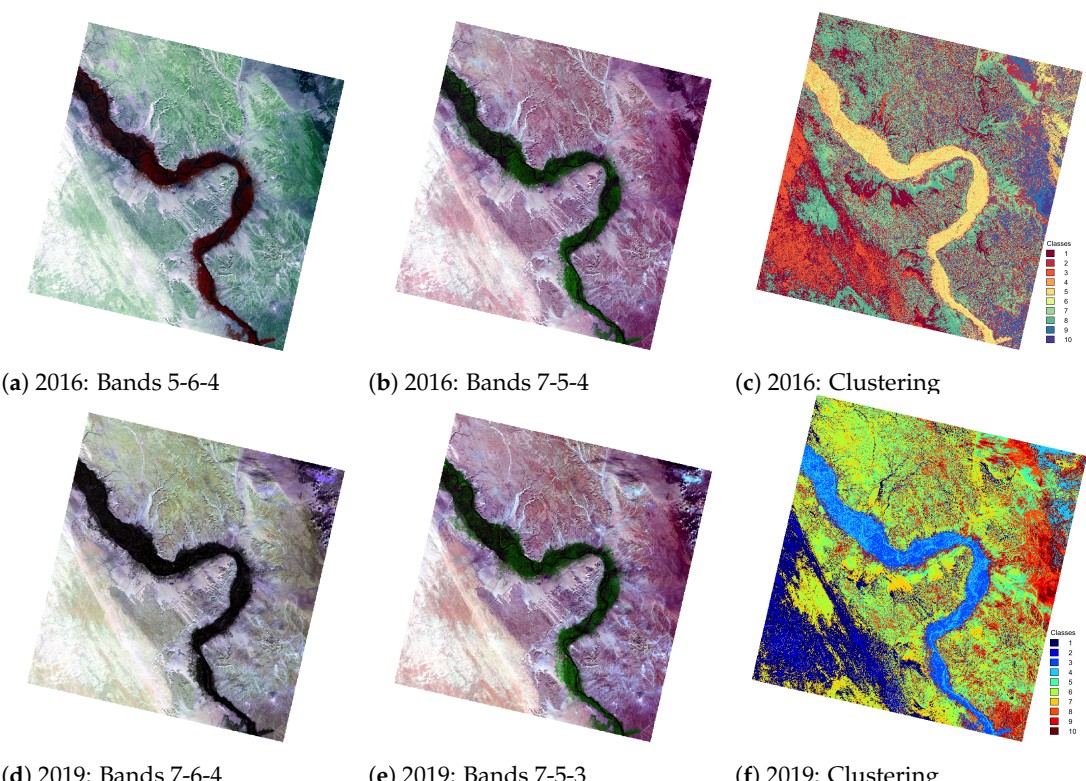

(**a**) 2016: Bands 5-6-4      (**b**) 2016: Bands 7-5-4      (**c**) 2016: Clustering

(**d**) 2019: Bands 7-6-4      (**e**) 2019: Bands 7-5-3      (**f**) 2019: Clustering

**Figure 8.** Qena Bend of the Nile River: false color composites and clustering of the Landsat 8-9 OLI/TIRS images: (**a**) 2016 RGB in 5-6-4 Bands, (**b**) 2016 RGB in 7-5-4 Bands, (**c**) 2016 unsupervised classification, (**d**) 2019 RGB in 7-6-4 Bands, (**e**) 2019 RGB in 7-5-3 Bands, (**f**) 2019 RGB: Clustering.

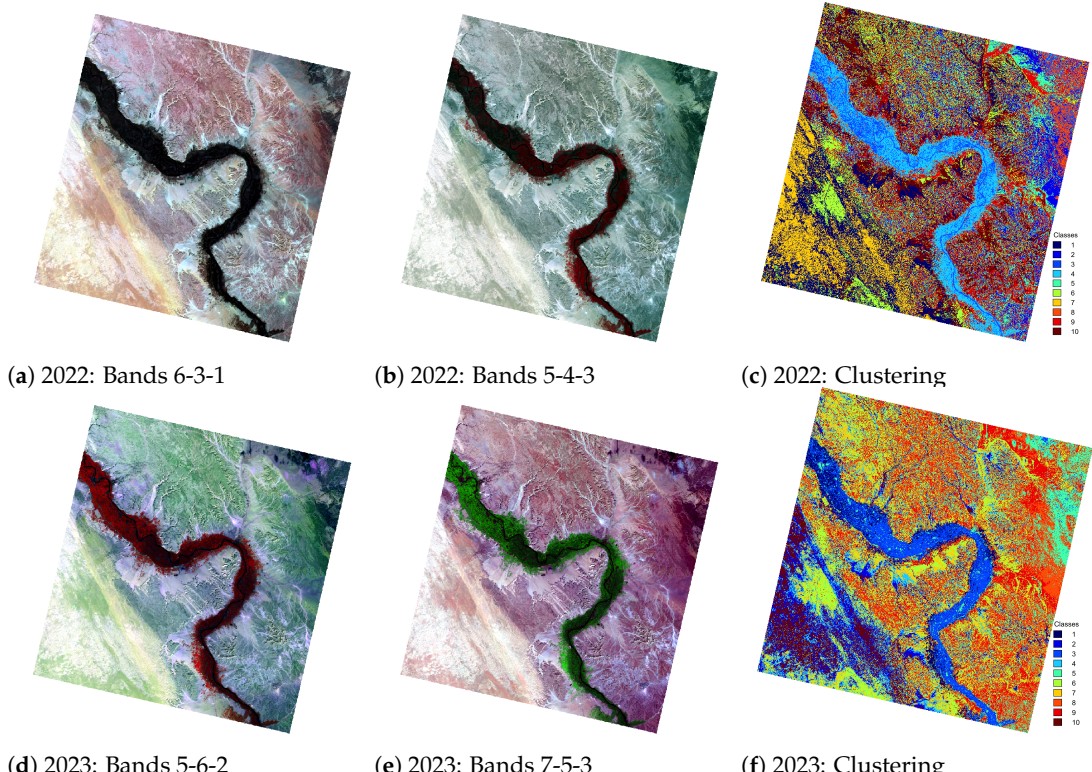

(**a**) 2022: Bands 6-3-1      (**b**) 2022: Bands 5-4-3      (**c**) 2022: Clustering

(**d**) 2023: Bands 5-6-2      (**e**) 2023: Bands 7-5-3      (**f**) 2023: Clustering

**Figure 9.** Qena Bend of the Nile River: false color composites and clustering of the Landsat 8-9 OLI/TIRS images: (**a**) 2022 RGB in 6-3-1 Bands, (**b**) 2022 RGB in 5-4-3 Bands, (**c**) 2022 unsupervised classification, (**d**) 2023 RGB in 5-6-2 Bands, (**e**) 2023 RGB in 7-5-3 Bands, (**f**) 2023 RGB: Clustering.

## 4. Results

In this paper, the classified land cover maps were plotted via the unsupervised classification of k-means clustering methods and the statistics of pixel assignments are summarized in Table 2. Using R, we have computed maps for the pairwise comparison of the different target years (2013 and 2015, 2016 and 2019, 2022 and 2023) to obtain a general trend of the transformation in selected land cover types in the Qena region. In the discrimination of land cover types, the spectral signatures of the satellite images are taken as markers to distinguish the landscape features based on their texture, shape, and size. In this case, color composites created based on the band combination of the Landsat images are included in the data processing and clustering.

**Table 2.** Results of the k-means classification of the Landsat-8 OLI images for Qena Bend area [1].

| Class ID | 2013 | 2015 | 2016 | 2019 | 2022 | 2023 |
|---|---|---|---|---|---|---|
| 1 | 6,237,758.6 | 25,058,427 | 11,830,859 | 11,050,520 | 33,773,365.2 | 22,414,794 |
| 2 | 26,530,972.6 | 3,829,403 | 6,847,520 | 7,687,698 | 21,467,288.5 | 4,080,038 |
| 3 | 46,356,528.0 | 23,160,871 | 13,056,437 | 31,925,587 | 466,472.5 | 13,918,989 |
| 4 | 10,314,256.8 | 7,458,695 | 12,594,525 | 3,359,878 | 3,547,592.0 | 1,863,680 |
| 5 | 7,861,914.8 | 9,909,609 | 3,012,708 | 11,101,016 | 7,714,730.1 | 6,083,803 |
| 6 | 625,035.3 | 4,568,955 | 2,195,275 | 14,290,386 | 9,346,294.4 | 14,743,313 |
| 7 | 22,694,195.3 | 3,019,950 | 12,277,811 | 14,898,378 | 6,978,872.8 | 6,370,890 |
| 8 | 5,804,917.1 | 1,942,973 | 10,144,633 | 5,386,949 | 16,906,933.2 | 6,053,497 |
| 9 | 7,320,502.4 | 13,450,745 | 3,220,846 | 5,501,193 | 8,429,786.2 | 11,952,344 |
| 10 | 6,121,676.5 | 10,228,437 | 4,195,904 | 10,594,134 | 11,161,798.4 | 10,517,903 |

[1] Sum of squares by clusters within computed cluster centroids.

The results of the clustering include the defined 10 classes best representing the land surface structure and corresponding to the following items modified and generalized from the existing study [108]: *(1) floodplain of the Nile River; (2) silt sediments; (3) wadi deposits; (4) gritty and gravel soil; (5) sandy soils; (6) fine sand, silty and clay soil; (7) gravel and stony soil; (8) terrace soils and stony debris; (9) limestone foothills; (10) limestone rock land*.

The information was retrieved from the analysis of clustering implemented on the Landsat satellite images that represents the multi-component structure of the Earth's surface and different types of landscapes. The Landsat 8-9 OLI/TIRS images were differentiated using clustering techniques, which enabled us to analyze the images in terms of the spatial distribution of pixels with different values according to their spectral signatures. The land cover types of the Qena Bend region were recognized automatically by R for the years 2013, 2015, 2016, 2019, 2022, and 2023. The dimensions of all the images are identical for all the Landsat scenes and correspond to 7801 rows, 7651 columns, and 59,685,451 cells (or pixels) with a resolution of 30 × 30 m, i.e., each pixel on the image corresponds to a 30 m patch size on the land surface. Thus, the number of pixels computed for each ID class gives the area of the land cover type with respective changes over the years. The statistics of pixels assigned to the land cover ID classes are based on the spectral signature of the DN of the pixels; see Table 2. Correspondingly, Tables 3–5 show the detected numbers of pixels for each land cover ID class in the Qena Bend region, for a quantitative pairwise comparison for the years 2013 and 2015, 2016 and 2019, and 2022 and 2023, respectively.

The land cover class 'floodplain' decreased by 5.5% from 2013 (ID class 2 with 26,530,972.6 pixels) to 2015 (ID class 1 with value 25,058,427), and then slightly increased at the end of the study period from 21,467,288.5 to 22,414,794 of the assigned pixels, i.e., by 4.22%; see Table 2. Land class 2 'silt sediments' showed a change from 2016 with 10,144,633 assigned pixels against 10,594,134 pixels in 2019, which demonstrates an increase of 4.24%. Land class 3 'wadi deposits' demonstrated the following changes in the distribution of pixels, from 6,237,758.6 in 2013 to 6,847,520 in 2015, i.e., an increase of 8.9%. From 2022 to 2023, the values changed from 6,978,872.8 to 6,053,497, i.e., 13.2%. A slight increase

in the same land cover class was further noted in 2022 with 11,161,798.4 assigned pixels, followed by a slight decrease to 10,517,903 in 2023, which resulted in a decline of 5.77%.

**Table 3.** Pairwise comparison of the k-means classification for Bands 5-4-3 of the Landsat-8 OLI/TIRS images for 2013 and 2015 in Qena Bend region [1].

| Class ID | 2013 | | | 2015 | | |
|---|---|---|---|---|---|---|
| | Band 5 | Band 4 | Band 3 | Band 5 | Band 4 | Band 3 |
| 1 | 21,019.08 | 18,348.92 | 15,722.58 | 19,819.50 | 17,226.143 | 14,765.14 |
| 2 | 21,956.38 | 19,148.12 | 16,294.50 | 12,087.00 | 10,917.333 | 10,115.00 |
| 3 | 26,892.38 | 23,105.00 | 19,120.92 | 18,736.17 | 9956.333 | 10,142.17 |
| 4 | 14,308.50 | 13,163.75 | 11,730.75 | 24,835.46 | 21,496.385 | 17,990.38 |
| 5 | 19,246.50 | 16,942.69 | 14,626.81 | 23,315.56 | 20,178.688 | 17,043.31 |
| 6 | 17,656.18 | 15,405.09 | 13,460.00 | 26,302.78 | 22,807.222 | 19,156.67 |
| 7 | 23,465.45 | 20,280.18 | 17,179.82 | 15,028.50 | 13,507.500 | 12,109.33 |
| 8 | 16,549.67 | 10,211.17 | 10,132.83 | 29,412.00 | 25,219.500 | 20,731.50 |
| 9 | 22,484.00 | 19,481.25 | 16,802.50 | 17,737.46 | 15,833.385 | 13,783.23 |
| 10 | 24,895.53 | 21,740.20 | 18,326.47 | 17,737.46 | 18,747.389 | 15,990.33 |

[1] Clusters are computed for each band in the composite triplet: Band 5 (NIR), Band 4 (Red), and Band 3 (Green).

**Table 4.** Pairwise comparison of the k-means classification for Bands 5-4-3 of the Landsat-8 OLI/TIRS images for 2016 and 2019 in Qena Bend region [1].

| Class ID | 2016 | | | 2019 | | |
|---|---|---|---|---|---|---|
| | Band 5 | Band 4 | Band 3 | Band 5 | Band 4 | Band 3 |
| 1 | 25,067.53 | 21,805.82 | 18,283.65 | 20,721.42 | 18,438.42 | 15,725.33 |
| 2 | 19,837.27 | 17,506.45 | 15,060.45 | 24,198.84 | 21,005.26 | 17,763.84 |
| 3 | 27,148.44 | 23,483.89 | 19,596.67 | 25,536.93 | 22,109.73 | 18,481.20 |
| 4 | 16,049.43 | 13,237.00 | 11,880.43 | 18,257.00 | 9376.75 | 9707.50 |
| 5 | 19,541.25 | 9323.50 | 9742.75 | 14,476.67 | 12,599.83 | 11,411.17 |
| 6 | 12,780.50 | 11,767.00 | 10,670.50 | 16,992.00 | 15,236.88 | 13,304.75 |
| 7 | 21,370.47 | 18,891.65 | 16,208.94 | 18,829.53 | 16,928.47 | 14,654.00 |
| 8 | 23,122.89 | 20,069.28 | 16,927.33 | 31,113.00 | 26,355.00 | 21,077.00 |
| 9 | 18,240.88 | 16,536.62 | 14,205.25 | 27,920.00 | 23,667.00 | 19,115.00 |
| 10 | 17,269.14 | 15,186.43 | 13,284.43 | 22,736.25 | 19,858.25 | 16,938.06 |

[1] Clusters are computed for each band in the composite triplet: Band 5 (NIR), Band 4 (Red), and Band 3 (Green).

**Table 5.** Pairwise comparison of the k-means classification for Bands 5-4-3 of the Landsat-8 OLI/TIRS images for 2022 and 2023 in Qena Bend region [1].

| Class ID | 2022 | | | 2023 | | |
|---|---|---|---|---|---|---|
| | Band 5 | Band 4 | Band 3 | Band 5 | Band 4 | Band 3 |
| 1 | 24,428.50 | 21,422.20 | 18,051.50 | 21,513.50 | 10,245.750 | 10,483.000 |
| 2 | 19,238.75 | 17,313.00 | 14,911.25 | 25,528.40 | 22,234.800 | 18,665.200 |
| 3 | 27,402.67 | 23,631.83 | 19,472.50 | 17,918.11 | 9096.444 | 9575.333 |
| 4 | 22,303.50 | 19,676.83 | 16,793.33 | 24,549.75 | 21,362.375 | 17,997.875 |
| 5 | 17,517.33 | 11,594.67 | 10,989.33 | 15,965.40 | 14,835.200 | 13,574.400 |
| 6 | 12,943.67 | 11,050.00 | 10,303.00 | 22,865.67 | 19,456.933 | 16,537.733 |
| 7 | 17,225.22 | 15,122.22 | 13,253.00 | 23,488.23 | 20,695.000 | 17,726.000 |
| 8 | 25,640.82 | 22,323.82 | 18,730.73 | 18,460.75 | 16,531.833 | 14,379.417 |
| 9 | 20,699.44 | 18,365.81 | 15,840.12 | 20,155.57 | 18,063.786 | 15,478.143 |
| 10 | 23,487.22 | 20,428.72 | 17,181.22 | 27,425.50 | 23,626.900 | 19,359.200 |

[1] Clusters are computed for each band in the composite triplet: Band 5 (NIR), Band 4 (Red), and Band 3 (Green).

A slight increase of 8.73% in land cover class 8 'terrace soils and stony debris' was noted, from 2015 with 3,829,403 pixels to 4,195,904 pixels in 2016. The land cover class 'limestone foothills' remained with comparable values in 2015 with 13,450,745 pixels against 12,277,811 in 2016, which is a decrease of 8.72 %, and 11,101,016 in 2019, i.e., 9.58%. Finally,

land cover class 10 'limestone rock land' experienced recent changes from 2022 with 8,429,786.2 values to 6,083,803 in 2023, which might be caused by the soil erosion processes. To compare the land cover types over Qena, we can start with Figure 7 and analyze changes in land cover classes and how the pixel classes are assigned, as summarized in Table 3.

Figure 7 shows several band combinations for color composites. False color composites (NIR, Red, and Green) were applied to better recognize the attributes of vegetation, water, and land as major classes via automatic machine-based algorithms of R. As a result, subclasses such as wadi valleys, the Nile floodplain and valley, and hills were defined according to the surface structure in the study area. The orientation of the patterns of Wadi Qena on the satellite images is almost parallel to the axis direction of the Red Sea uplift, which corresponds to the geological origin of the wadi network. Notably, Band 4-3-2 represents a natural color composite; Band 5-4-3 is a composition that includes an infrared (Band 5) and is therefore adjusted for the identification of vegetation in the river floodplain.

Band 5-7-1 contains NIR (Band 5), SWIR2 (Band 7), sensitive to radiation emission, and coastal aerosol (Band 1), sensitive to small particles, haze, and also burnt areas [109]. The area of the Nile floodplain with riparian vegetation has strong absorption both in the visible and NIR bands, due to the pigmentation of chlorophyll, which absorbs highly at wavelengths in visible bands. Other factors are the physiological structure of the plant canopy and the moisture of leaves. Therefore, the floodplain area has strong reflectance in all bands due to the presence of vegetation with high water content, as can be seen in Figures 7–9.

A pairwise comparison of the land cover changes enables us to obtain a general picture of the gradual trends in land cover changes in Qena's surroundings over the past 10 years. Figures 7–9 present a pairwise comparison of the changes in land cover types in the Qena Bend region up to the relevant two years next to each other. The analysis of these images demonstrates maps of changes in land cover types calculated and visualized between the years 2013 and 2015 (Figure 7), the years 2016 and 2019 (Figure 8), and the years 2022 and 2023 (Figure 9). The computation is summarized in Table 3 for 2013/2015, Table 4 for 2016/2019, and Table 5 for 2022/2023.

Figure 8 demonstrates the results of the clustering for the pairwise comparison of years 2016 and 2019 and the color composites using the following band combinations: 5-6-4; 7-5-4, 7-6-4 and 7-5-3. False color composites 7-5-4 and 7-6-4 (urban) include the Shortwave Infrared band (SWIR-2), which is optimal for the visualization of urban areas due to better contrast between the built-up areas, water areas, and vegetation areas. Moreover, while composites 7-5-4 and 7-5-3 resemble each other due to the presence of the SWIR1 and Red bands, the Green band (Band 3) in the second composite enables the better discrimination of the limestone rock land using a bright blue color, whereas, in composite 7-5-4, the color used for representing limestone rock land is very similar to that used for the gravel and stony soil, namely orchid and magenta, respectively.

Furthermore, the area covered by land cover class 4 'gritty and gravel soil' remained stable, with only a very slight decrease in the area from 2015 with 3,019,950 assigned pixels to 2016 with 3,012,708 pixels, which is below 1%. Moreover, land cover class 5 'sandy soils' demonstrated an increase in recent years from 11,161,798.4 to 11,952,344, i.e., an increase of 6.6 %. Land cover class 6 'fine sand, silty and clay soil' has changes, with a slight increase from 3,220,846 in 2016 to 3,359,878 in 2019, which is a gain of 4.14%, which could be caused by the environmental effects related to the hydrology of the Nile. Land cover class 7 'gravel and stony soil' showed a slight decrease from 7,687,698 in 2019 to 7,714,730.1 in 2022, which is below 1%.

Figures 7–9 show the wadi streams with their typical dendrite structure, with a comparison of images for the years 2013, 2015, 2016, 2019, 2022, and 2023. The fluvial network structure is oriented towards the Qena Bend segment of the Nile. The system of the wadi fluvial structure and drainage stream network in the surroundings of the Qena Bend of the Nile River was identified in the images due to its characteristic morphological structure with a typical dendritic pattern that represents the flow direction of the dry wadi

streams. These are largely controlled by the underlying geological structure and regional lithology. The origin of these basins lies in the mountainous highland of the basement and is oriented west–east, either to the Nile or to the Red Sea, according to the catchment, which means that their direction follows these orientations. In both cases, the potential damage from disastrous events may affect urban areas such as the small cities and the infrastructure of industrial locations.

Figure 9 visualizes the band combinations 6-3-1, 5-4-3, 5-6-2, and 7-5-3 for the images covering the Qena region in 2022 and 2023. The composite of 5-4-3 includes the NIR, Red, and Green bands that best represent the vegetation coverage in the Nile River floodplain, which is identified as dark crimson, well contrasted against the other land cover types. The composite 5-6-2 includes the SWIR 1 and Blue bands, which results in the sands of the Western and Eastern Deserts shown in green shadows. This enables us to discriminate them from the wadi channels that are colored in light lilac. The limestone foothills are visualized in light magenta, while gritty and gravel soil appears slate blue. Sandy soils are colored in light chartreuse green, while the terrace soils and stony debris are in lime green.

## 5. Discussion

The Qena Wadi region is affected by annual flash floods and related geomorphological processes such as erosion as a consequence of the soil's mass movement and the deposition of soil particles during the heavy rainfall. In turn, the deterioration of the soil and changes in land cover types have a negative impact on agriculture and may damage infrastructure, e.g., cut off roads. The consequences of such natural hazards emphasize the dependence of social sustainability and urban infrastructure on the environmental setting of the region. Accurate information regarding the possibility of geomorphic risks or landslide deposition may be derived from an evaluation of the landscape's morphology. For instance, the evaluation of the channel incision modeled by geomorphic adjustment scenarios can demonstrate the effects of geomorphic processes on infrastructure [110].

Our study aimed to visualize gradual land cover changes in Upper Egypt and illustrate the geomorphology and topography of the Wadi Qena region. To achieve this, we employed a combination of methods, including the classification of time series of satellite images, extraction of 2D and 3D mapping information from topographic maps (GEBCO/SRTM grids), and evaluation of changes in land cover types using statistical data derived from image analysis. Computer vision algorithms for image recognition, discrimination, and data processing are fundamental problems in remote sensing and Earth sciences, with a wide range of applications [111–115]. The use of computer vision algorithms for the visualization, analysis, and classification of satellite images aims to prevent the risk of natural hazards by mapping potentially endangered areas. Regions such as wadi valleys in the desert areas of Southern Egypt are at high risk of flash floods.

To recognize changes in the satellite images, short time series analysis was performed using the principles of remote sensing data processing [116]. Similarity in the texture and structure of the objects represented on the Landsat scenes was analyzed based on the characteristics of the multispectral channels from the Landsat satellite images using fusion techniques for diverse spectral bands [117,118]. Flash floods and soil deposition in the Wadi Qena region are natural phenomena related to the climate and occur repeatedly in Southern Egypt due to its specific geomorphology [119]. Therefore, predicting such hazards requires complex climatic computational modeling. However, robust information derived from remote sensing data processed by advanced computer vision and pattern recognition tools can support risk mitigation measures and propose planning strategies in regions prone to natural hazards. Geomorphic data can also be used to evaluate groundwater supply [120,121] and its availability for irrigation in agriculture [122].

Therefore, satellite-derived maps of land cover types are a useful tool for both determining hazard areas and defining zones susceptible to floods, and we also applied geomorphic modeling according to the distribution of the fluvial network of the wadi in the southern segment of the Eastern Desert. Moreover, data analysis and visualization

performed using scripting methods provide a new source of information that can be used to implement mitigation policies and measures of sustainable development in the Qena Bend region, as well as for urban planning in hazard-prone areas of Egypt [123].

In future works, we plan to assess the geomorphological hazards by analyzing the susceptibility of various regions supported by the classified satellite images. This will enable the evaluation of the risk of flash floods, soil erosion, and mass deposition. Since landscapes are subject to destructive geomorphological hazards such as flash floods, dust storms, and heavy showers, the use of satellite images provides objective and robust information regarding these hazards and their disastrous processes. Flash floods, in particular, cause sediment deposition and debris and affect the shape of the land surface. Therefore, the preventive mapping and evaluation of geomorphological risks are crucial for infrastructure and agricultural activities in Upper Egypt.

## 6. Conclusions

The use of satellite images in the classification of land cover types by clustering is an important approach for applications in Earth sciences. At the same time, identification of the features and properties of the landscapes on the space-borne images is a challenging field of investigation in geoinformation due to the complexity of the interpretation of the land cover classes. A multi-disciplinary approach combining remote sensing data and advanced technical tools for their processing is an effective approach that supports satellite image analysis. In this paper, we demonstrate the application of such an approach by presenting the programming tools used for spatial data analysis. We show that the advanced methods of scripting languages for image processing are effective tools for image analysis and classification.

Specifically, we have demonstrated a method for extracting information from the Landsat 8-9 OLI/TIRS satellite images to track land cover changes in the wadi drainage surface of the Qena Bend district of the Nile River. We have presented an application of the R scripting libraries for the processing of geoinformation to analyze visual objects on the remote sensing data by computer vision and methods of pattern recognition on the images. Local features in the target study area were identified using the k-means clustering method and visualized in synthesized maps based on the six classified images. This algorithm exploits the values of the pixels' DNs to discriminate the classes on the land surface based on the similarity of pixels' values and assigning them to 10 separate classes.

We have approached the problem of unsupervised classification of the satellite images through the use of the R language, to extract objects from information on the land cover types in the Qena Bend region in Upper Egypt. The R scripting libraries 'raster', 'terra', and 'RStoolbox' were utilized to extract and analyze information from remote sensing data in the context of the geomorphological structures over the studied area. Through a machine-based approach enabled by R, spatial features and land cover classes were automatically separated, allowing for the analysis of changes in the ephemeral riverbed complex in the Qena district using computer vision and applied programming.

In the framework of this study, the scripted implementation of the R libraries was evaluated under Landsat OLI/TIRS data with auxiliary cartographic data processing by GMT. The comparison of the land cover changes was then summarized in tabular form, based on information extracted from the R-processed images. Our analyses show that the R approach algorithms of computer vision applied for geographic data processing are effective for multispectral space-borne data.

Our study demonstrates the effectiveness of the R-based method as an advanced approach to geospatial image processing compared to state-of-the-art GIS methods. Specifically, our algorithm is trained to classify pixels and assign them to different land cover classes based on the spectral reflectance values in various bands of the satellite images, allowing for automatic interpretation of the images. The use of computer vision algorithms and automation allowed for the generation of a series of maps depicting the land cover types in the Qena area during the years 2013, 2015, 2016, 2019, 2022, and 2023. Comparing

the changes in land cover types provides a better understanding and interpretation of the cumulative effects of environmental changes and occasional floods in the southern segment of the Eastern Desert on the geomorphic patterns of land cover types in the Qena Bend region of Upper Egypt.

**Author Contributions:** Supervision, conceptualization, methodology, software, resources, funding acquisition, and project administration, O.D.; writing—original draft preparation, methodology, software, data curation, visualization, formal analysis, validation, writing—review and editing, and investigation, P.L. All authors have read and agreed to the published version of the manuscript.

**Funding:** The publication was funded by the Editorial Office of *Information*, MDPI, by providing a 100% discount for the APC of this manuscript. This project was supported by the Federal Public Planning Service Science Policy or Belgian Federal Science Policy Office—BELSPO (B2/202/P2/SEISMOSTORM).

**Institutional Review Board Statement:** Not applicable.

**Informed Consent Statement:** Not applicable.

**Data Availability Statement:** Not applicable

**Acknowledgments:** The authors thank the reviewers for their reading and review of this manuscript.

**Conflicts of Interest:** The authors declare no conflict of interest.

## Abbreviations

The following abbreviations are used in this manuscript:

| | |
|---|---|
| ASTER | Advanced Spaceborne Thermal Emission and Reflection Radiometer |
| DBSCAN | Density-Based Spatial Clustering of Applications with Noise |
| GEBCO | General Bathymetric Chart of the Oceans |
| GMT | Generic Mapping Tools |
| GIS | Geographic Information System |
| Landsat OLI/TIRS | Landsat Operational Land Imager Thermal Infrared Sensor |
| NIR | Near Infrared |
| RGB | Red Green Blue |
| SRTM | Shuttle Radar Topography Mission |
| SWIR | Shortwave Infrared |
| DEM | Digital Elevation Model |
| UTM | Universal Transverse Mercator |
| WGS84 | World Geodetic System 1984 |

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
