# Peer review of "Recognizing the Wadi Fluvial Structure and Stream Network in the Qena Bend of the Nile River, Egypt, on Landsat 8-9 OLI Images"

_information, doi:10.3390/info14040249_

Round 1

Reviewer 1 Report

The paper discusses the unsupervised classification of Landsat OLI/TIRS images using R for visualizing land surface structures in the Qena governorate of Upper Egypt. The study focuses on identifying the fluvial structure and stream network of Wadi Qena region by analysing high-resolution 2D and 3D topographic and geologic maps. The research paper is generally well-presented, but can be improved in the following aspects:

1.       The related works need to be enhanced. More relevant and recent research works should be added.

2.       The motivations should be highlighted. The paper proposed an R-based case of satellite image processing to address the problem of computer vision applications. Why an R-based case is proposed needs to be clarified. More clarifications regarding the differences between this research work and others are recommended.

3.       The choice of techniques needs to be explained. The classified land cover maps have been plotted through k-means clustering methods. Why not other clustering algorithms? Furthermore, how is the number of clusters identified? 

Author Response

Dear Editors of Information,

We are pleased to resubmit our revised manuscript.

Please find attached the revised version of the paper. We have carefully followed all the comments and suggestions of the reviewers and corrected the manuscript accordingly.

All the corrections in the text are marked up yellow for Track Changes. The proofreading of the manuscript has been done by the native speaker colleague (UK citizen).

The replies to the comments of the reviewers are listed below.

Using the opportunity, we thank the reviewers for careful reading of the paper which improved the initial version of the manuscript.

With kind regards, - Authors (Polina Lemenkova and Olivier Debeir).

11.04.2023.

Reviewer 1

No

Reviewer’s Comments

Author’s actions

1

Is the research design appropriate? – Can be improved.

The research design is improved in section 3. Materials and Methods; subsection 3.2. Methods. All the updates are coloured yellow. Also, the proofreading of the whole manuscript is done by the native speaker colleague and the text is updated.

2

Are the methods adequately described? – Can be improved.

The Section 3.2. Methods is restructured into the 2 sub-subsections according to the logical flow of the workflow. The concept and advantages of the research method is presented in the first part, while the data processing flow in the next one. A comparative analysis summarizing the clustering results in the study area is added. The reasons for the selection of different image time intervals are explained.

3

Are the results clearly presented? – Can be improved.

The data analysis is made in more details. Thus, the results of the clustering of each image obtained are further analyzed and described. In particular, more comments are added in the section 4. Results on Figures 7, 8 and 9 as well as Tables 2 to 5.

4

The paper discusses the unsupervised classification of Landsat OLI/TIRS images using R for visualizing land surface structures in the Qena governorate of Upper Egypt. The study focuses on identifying the fluvial structure and stream network of Wadi Qena region by analysing high-resolution 2D and 3D topographic and geologic maps. The research paper is generally well-presented, but can be improved in the following aspects: The related works need to be enhanced. More relevant and recent research works should be added.

The Introduction section is restructured and updated. Two new subsections are now included: 1.1. Background and 1.2. Motivation and Objectives. The related works are enhanced with cited more works (over 10 studies) as suggested. These illustrate the use of diverse methods of GIS for processing satellite images and obtaining information from remote sensing data. New paragraphs are inserted in the Introduction section with a review of the relevant works on the use of Landsat / Sentinel satellite images for mapping Egypt. More relevant and recent research works are cited and discussed. Their limitations are also discussed in comparison of this work.

5

The motivations should be highlighted. The paper proposed an R-based case of satellite image processing to address the problem of computer vision applications. Why an R-based case is proposed needs to be clarified. More clarifications regarding the differences between this research work and others are recommended.

The new subsection is added in the Introduction: 1.2. Motivation and Objectives. In this subsection, several paragraphs on the advances in remote sensing data processing are added explaining the motivation to use R programming language instead of traditional GIS approaches. Examples of the processing satellite images by the traditional GIS are also included and the applications explained. Moreover, some more information is added highlighting the differences between our research work and previous studies that also used remote sensing data for mapping landscapes of Egypt.

6

The choice of techniques needs to be explained. The classified land cover maps have been plotted through k-means clustering methods. Why not other clustering algorithms? Furthermore, how is the number of clusters identified?

The research design is updated in the Methods section added more explanation on the techniques and clustering research approach. The details of the k-means clustering methods used for classification of the land cover maps are provided.

Original review report

Open Review

Quality of English Language

( ) English very difficult to understand/incomprehensible
( ) Extensive editing of English language and style required
( ) Moderate English changes required
(x) English language and style are fine/minor spell check required
( ) I am not qualified to assess the quality of English in this paper

Yes

Can be improved

Must be improved

Not applicable

Does the introduction provide sufficient background and include all relevant references?

(x)

( )

( )

( )

Are all the cited references relevant to the research?

(x)

( )

( )

( )

Is the research design appropriate?

( )

(x)

( )

( )

Are the methods adequately described?

( )

(x)

( )

( )

Are the results clearly presented?

( )

(x)

( )

( )

Are the conclusions supported by the results?

(x)

( )

( )

( )

Comments and Suggestions for Authors

The paper discusses the unsupervised classification of Landsat OLI/TIRS images using R for visualizing land surface structures in the Qena governorate of Upper Egypt. The study focuses on identifying the fluvial structure and stream network of Wadi Qena region by analysing high-resolution 2D and 3D topographic and geologic maps. The research paper is generally well-presented, but can be improved in the following aspects:

1. The related works need to be enhanced. More relevant and recent research works should be added.

2. The motivations should be highlighted. The paper proposed an R-based case of satellite image processing to address the problem of computer vision applications. Why an R-based case is proposed needs to be clarified. More clarifications regarding the differences between this research work and others are recommended.

3. The choice of techniques needs to be explained. The classified land cover maps have been plotted through k-means clustering methods. Why not other clustering algorithms? Furthermore, how is the number of clusters identified?

Submission Date

06 March 2023

Date of this review

14 Mar 2023 03:22:19

Reviewer 2 Report

The references can be shorted and the Tables can be reformated.  

Author Response

Dear Editors of Information,

Please find attached the revised version of the paper. We have carefully followed all the comments and suggestions of the reviewers and corrected the manuscript accordingly.

All the corrections in the text are marked up yellow for Track Changes. The proofreading of the manuscript has been done by the native speaker colleague (UK citizen). The English language of the manuscript is checked throughout. All the occasional typesetting misprints and minor grammar mistakes (spelling, punctuation, syntax) were corrected where necessary.

The replies to the comments of the reviewers are listed below.

Using the opportunity, we thank the reviewers for careful reading of the paper which improved the initial version of the manuscript.

With kind regards, - Authors (Polina Lemenkova and Olivier Debeir).

12.04.2023.

Reviewer 2

No

Reviewer’s Comments

Author’s actions

1

The references can be shorted and

The Introduction section was restructured according to the two more references. Therefore, the references are updated and partially changed.

2

The Tables can be reformatted.

The Results section containing Tables is updated with included more new paragraphs and detailed comments on the results of the image clustering. The comparison of the images is enlarged and extended with added more comments on the variation of pixels assigned to different types of land cover classes. The tables are discussed in added paragraphs with regard to the land cover classes.

The changes of each land cover class were examined and compared pairwise over years with evaluated changes compared in Tables. Analysing the changes up to each two years next to each other (e.g., 2013 to 2015, 2016 to 2019, 2022 to 2023), we also commented on the overall changes in the land cover types based on the results of clustering in pictures over the last 10-year period (2013-2023). In particular, the following paragraphs are inserted starting with “The information was retrieved from the analysis of clustering implemented on the Landsat satellite images that represents the multi-component structure of the surface <...>”; “he land cover class 'floodplain' decreased on 5.5% <...>” and “A slight increase in 8.73% in the land cover class ‘terrace soils and stony debris’”; “Furthermore, the area covered by the land cover class 4 ’gritty and gravel soil’ <...>”.

3

Does the introduction provide sufficient background and include all relevant references? – Yes

Are all the cited references relevant to the research? – Yes

Is the research design appropriate? – Yes

Are the methods adequately described? – Yes

Are the results clearly presented? – Yes

Are the conclusions supported by the results? – Yes

Many thanks for the overall positive evaluation of the manuscript and support. The paper is updated and partially restructured, according to the two more reviewer’s reports where new paragraphs, sections and sentences are all introduced and coloured yellow for track changes. The paper is also proofread by the native speaker and improved in context. More information is added on the concepts of K-means clustering, research design is updated with included new sub-subsections in the Methods section. All changes are marked up in the text.

Original review report

Open Review

Quality of English Language

( ) English very difficult to understand/incomprehensible
( ) Extensive editing of English language and style required
( ) Moderate English changes required
(x) English language and style are fine/minor spell check required
( ) I am not qualified to assess the quality of English in this paper

Yes

Can be improved

Must be improved

Not applicable

Does the introduction provide sufficient background and include all relevant references?

(x)

( )

( )

( )

Are all the cited references relevant to the research?

(x)

( )

( )

( )

Is the research design appropriate?

(x)

( )

( )

( )

Are the methods adequately described?

(x)

( )

( )

( )

Are the results clearly presented?

(x)

( )

( )

( )

Are the conclusions supported by the results?

(x)

( )

( )

( )

Comments and Suggestions for Authors

The references can be shorted and the Tables can be reformated.  

Submission Date

06 March 2023

Date of this review

12 Apr 2023 17:15:58

Reviewer 3 Report

The overall content of the article is concise, six images were processed using methods of computer vision by R libraries. The results of the k-means clustering of each scene retrieved from the multi-temporal images were compared to visualize changes in landforms caused by the cumulative effects from the geomorphological disasters and climate-environmental processes. The proposed methods of R scripts run effectively and performs favorably in computer vision issues aimed at geospatial image processing and analysis of remote sensing data. However, the article has the following shortcomings.

1.       The article focuses on the method as a whole, and the data analysis is not detailed enough.

2.       The summary section can be appropriately added to the comparative summary of the clustering results in the study area.

3.       The six image years in Figure 5 are inconsistent with the multi-temporal image data years proposed in the study. In addition, whether the selection of different image time intervals affects the comparison of clustering results needs further explanation.

4.       In Section 3.2, the concept and advantages description of the research method of the article should be distinguished from the data processing flow.

5.       Table 2, table 3, table 4 and table 5 should be further analyzed, and the data in the table are not fully described.

6.       The results of the clustering of each phase obtained by Figure 7, Figure 8 and Figure 9 should be further analyzed and described, what problems are explained, what results are obtained, and so on. At the same time, it is necessary to compare and analyze the clustering results of different phases in detail.

7.       The conclusion part should summarize the comparison of clustering results.

Author Response

Dear Editors of Information,

Please find attached the revised version of the paper. We have carefully followed all the comments and suggestions of the reviewers and corrected the manuscript accordingly.

All the corrections in the text are marked up yellow for Track Changes. The proofreading of the manuscript has been done by the native speaker colleague (UK citizen). The English language of the manuscript is checked throughout. All the occasional typesetting misprints and minor grammar mistakes (spelling, punctuation, syntax) were corrected where necessary.

The replies to the comments of the reviewers are listed below.

Using the opportunity, we thank the reviewers for careful reading of the paper which improved the initial version of the manuscript.

With kind regards, - Authors (Polina Lemenkova and Olivier Debeir).

12.04.2023.

Reviewer 3

No

Reviewer’s Comments

Author’s actions

1.

Extensive editing of English language and style required

The manuscript is proofread throughout by the native speaker colleague (citizen of the UK – Great Britain). All the occasional typesetting misprints and minor grammar mistakes are corrected where necessary: spelling, punctuation, syntax. Misprints are checked and corrected everywhere in the text.

2

Does the introduction provide sufficient background and include all relevant references? – Can be improved.

The Introduction section is improved and partially restructured. We divided it into the two new subsections: 1.1. Background and 1.2. Motivation and Objectives. The whole text is proofread and the logical flow of the text is checked. The motivation and objectives of this work are explained in the relevant subsection by pointing at the advantages of R programming approaches compared to the conventional methods, such as automation achieved by scripts, accuracy of image processing and machine-based data analysis. New paragraphs are inserted in the Introduction with an overview of the use of satellite images for environmental mapping of Egypt.

Moreover, additional references are included in the sub-subsection 3.2.1. Research concept and advantages, where various types of clustering are discussed, and explained the advantages of K-means techniques by R language.

3

Are all the cited references relevant to the research? – Can be improved.

Several new references are included in the Bibliography and cited in the Introduction where the overview of the existing works is presented and discussed in more details. New references are provided in the paragraphs of the Introduction started by “In view of the advantages of the satellite images for Earth and environmental studies <...>” and “Other studies have used Erdas Imagine for processing <...>”. The related works are included with cited several more works. New citations report the use of existing methods of GIS for satellite images analysis. Their limitations are also discussed in comparison of this work.

4

Are the methods adequately described? – Can be improved.

The Methodology section is updated, proofread and improved in context. More information is added on the concepts of K-means clustering, research design is updated with included new sub-subsections in the Methods section (pp. 8-9).

5

Are the results clearly presented? – Can be improved.

The Results section is updated with added more new paragraphs and comments on the results of image clustering. The comparison of the images is enhanced with more comments provided on the variation of pixels assigned to different types of land cover classes which enabled to detect changes in the landscapes of Qena from 2013 to 2022. the comparison of the land cover classes is done and commented. New paragraphs are also added on the changes in vegetation coverage agains the other land cover types in the studied landscapes such as 1) Nile River; 2) Floodplain with silts; 3) Wadi deposits; 4) Gritty and gravel soil; 5) Sand soils; 6) Fine sand, silty and clay soil; 7) Gravel and stony soil; 8) Terrace soils and stony debris; 9) Limestone foothills; 10) Limestone rock land. Each class was examined over years with evaluated changes compared by tables. Analysing the changes up to each two years next to each other (e.g., 2013 to 2015, 2016 to 2019, 2022 to 2023), we also commented on the overall changes in the land cover types based on the results of clustering in pictures over the last 10-year period (2013-2023).

6

The overall content of the article is concise, six images were processed using methods of computer vision by R libraries. The results of the k-means clustering of each scene retrieved from the multi-temporal images were compared to visualize changes in landforms caused by the cumulative effects from the geomorphological disasters and climate-environmental processes. The proposed methods of R scripts run effectively and performs favorably in computer vision issues aimed at geospatial image processing and analysis of remote sensing data. However, the article has the following shortcomings. The article focuses on the method as a whole, and the data analysis is not detailed enough.

Data analysis is enhanced with included more description and comments on the graphs and tables in the Results section. All the new paragraphs and phrases are coloured yellow for track changes. In particular, the following paragraphs are inserted starting with “The information was retrieved from the analysis of clustering implemented on the Landsat satellite images that represents the multi-component structure of the surface <...>”; “he land cover class 'floodplain' decreased on 5.5% <...>” and “A slight increase in 8.73% in the land cover class ‘terrace soils and stony debris’”; “Furthermore, the area covered by the land cover class 4 ’gritty and gravel soil’ <...>”.

7

The summary section can be appropriately added to the comparative summary of the clustering results in the study area.

The clustering results are compared in the section 4. Results in the following paragraphs: “The land cover class ’floodplain’ decreased on 5.5% from 2013 (ID class 2 with 26530972.6 pixels) to 2015 (ID class 1 with value 25058427), then <...>”; “A slight increase in 8.73% in the land cover class 8 ’terrace soils and stony debris’ was noted from 2015 with pixels to 4195904 pixels in 2016 <...>”; “Furthermore, the area covered by the land cover class 4 ’gritty and gravel soil’ remained stable with only a very slight decrease in the area from 2015 with 3019950 assigned pixels to 2016 with <...>”.

8

The six image years in Figure 5 are inconsistent with the multi-temporal image data years proposed in the study. In addition, whether the selection of different image time intervals affects the comparison of clustering results needs further explanation.

This information is added in the subsection 3.1. Data starting with phrase “The selection of different image time intervals is explained by <...>” and then followed by “Given the advantages of the Landsat 8-9 OLI/TIRS sensor <...>”. Two extended paragraphs are inserted explaining the choice of the Landsat OLI/TIRS data and period with time span.

9

In Section 3.2, the concept and advantages description of the research method of the article should be distinguished from the data processing flow.

Corrected as suggested: Section 3.2. Methods is divided into the 2 sub-subsections with relevant titles: 3.2.1. Research concept and advantages (a new sub-subsection with all text and citations included) and 3.2.2. Data processing workflow (updated, proofread and partially rewritten). The concept of clustering is discussed and its advantages are mentioned.

10

Table 2, table 3, table 4 and table 5 should be further analyzed, and the data in the table are not fully described.

Added three new paragraphs in pp. 11 and 13 with numerical analysis of the computed land cover classes by years. The percentage in variations is calculated. All new insertions are coloured yellow.

11

The results of the clustering of each phase obtained by Figure 7, Figure 8 and Figure 9 should be further analyzed and described, what problems are explained, what results are obtained, and so on. At the same time, it is necessary to compare and analyze the clustering results of different phases in detail.

The results are analysed more detailed with added comments on changes of each land cover classes. The new insertions are marked up for track changes. The computations are presented in pixels assigned by classes and by percentage of variations of these pixels by the time spans used for comparison.

12

The conclusion part should summarize the comparison of clustering results.

The Conclusion and Discussion sections are updated, partially restructured and improved. Added the closing remarks with the following paragraphs: “The use of satellite images in the classification of land cover types by clustering <...>”; “The R scripting libraries ’raster’, ’terra’, and ’RStoolbox’ were utilized to extract and analyse information from remote sensing data in the context of the geomorphological structures <...>”; “A pairwise comparison of the land cover changes enables to make an general <...>”; “Our study aimed to visualise gradual land cover changes in Upper Egypt <...>”.

Original review report

Open Review

Quality of English Language

( ) English very difficult to understand/incomprehensible
(x) Extensive editing of English language and style required
( ) Moderate English changes required
( ) English language and style are fine/minor spell check required
( ) I am not qualified to assess the quality of English in this paper

Yes

Can be improved

Must be improved

Not applicable

Does the introduction provide sufficient background and include all relevant references?

( )

(x)

( )

( )

Are all the cited references relevant to the research?

( )

(x)

( )

( )

Is the research design appropriate?

(x)

( )

( )

( )

Are the methods adequately described?

( )

(x)

( )

( )

Are the results clearly presented?

( )

(x)

( )

( )

Are the conclusions supported by the results?

(x)

( )

( )

( )

Comments and Suggestions for Authors

The overall content of the article is concise, six images were processed using methods of computer vision by R libraries. The results of the k-means clustering of each scene retrieved from the multi-temporal images were compared to visualize changes in landforms caused by the cumulative effects from the geomorphological disasters and climate-environmental processes. The proposed methods of R scripts run effectively and performs favorably in computer vision issues aimed at geospatial image processing and analysis of remote sensing data. However, the article has the following shortcomings.

1. The article focuses on the method as a whole, and the data analysis is not detailed enough.

2. The summary section can be appropriately added to the comparative summary of the clustering results in the study area.

3. The six image years in Figure 5 are inconsistent with the multi-temporal image data years proposed in the study. In addition, whether the selection of different image time intervals affects the comparison of clustering results needs further explanation.

4. In Section 3.2, the concept and advantages description of the research method of the article should be distinguished from the data processing flow.

5. Table 2, table 3, table 4 and table 5 should be further analyzed, and the data in the table are not fully described.

6. The results of the clustering of each phase obtained by Figure 7, Figure 8 and Figure 9 should be further analyzed and described, what problems are explained, what results are obtained, and so on. At the same time, it is necessary to compare and analyze the clustering results of different phases in detail.

7. The conclusion part should summarize the comparison of clustering results

Submission Date

06 March 2023

Date of this review

05 Apr 2023 07:35:32

Round 2

Reviewer 1 Report

The authors have addressed all my questions.